# Preparation and Performance of Supercritical Carbon Dioxide Thickener

**DOI:** 10.3390/polym13010078

**Published:** 2020-12-28

**Authors:** Bin Liu, Yanling Wang, Lei Liang

**Affiliations:** School of Petroleum Engineering, China University of Petroleum (East China), Qingdao 266580, China; B19020038@s.upc.edu.cn (B.L.); jerryleon1224@outlook.com (L.L.)

**Keywords:** supercritical carbon dioxide, thickener, molecular simulation, core damage, greenhouse effect

## Abstract

The low sand-carrying problem caused by the low viscosity of supercritical carbon dioxide (SC–CO_2_) limits the development of supercritical CO_2_ fracturing technology. In this study, a molecular simulation method was used to design a fluorine-free solvent-free SC–CO_2_ thickener 1,3,5,7-tetramethylcyclotetrasiloxane (HBD). Simulations and experiments mutually confirm that HBD-1 and HBD-2 have excellent solubility in SC–CO_2_. The apparent viscosity of SC–CO_2_ after thickening was evaluated with a self-designed and assembled capillary viscometer. The results show that when the concentration of HBD-2 is 5 wt.% (305.15 K, 10 MPa), the viscosity of SC–CO_2_ increases to 4.48 mPa·s. Combined with the capillary viscometer and core displacement device, the low damage of SC–CO_2_ fracturing fluid to the formation was studied. This work solves the pollution problems of fluoropolymers and co-solvents to organisms and the environment and provides new ideas for the molecular design and research of SC–CO_2_ thickeners.

## 1. Introduction

In shale gas hydraulic fracturing, water is an important resource. However, with the continuous application of hydraulic fracturing technology, problems such as water scarcity, drinking water pollution, and flowback water treatment have gradually become prominent [1]. Researchers began to focus on supercritical carbon dioxide (SC–CO_2_) fracturing fluid technology. Richard S. [1,2] systematically studied the pros and cons of using CO_2_ as the working fluid for shale gas production: (1) CO_2_ fracturing effect and fracture expansion are better than water-based hydraulic fracturing, and under constant pressure test conditions on the shale surface CO_2_ adsorption is better than CH_4_ at elevated temperature, which facilitates gas extraction and also serves as a fixation of CO_2_ [3] show in Figure 1, (2) Possible shortcomings, including the cost and safety issues related to handling a large amount of SC–CO_2_: separation of CO_2_ and CH_4_ mixed gas, transportation costs, pressure safety, and other issues. SC–CO_2_ has the characteristics of high diffusion coefficient, ultra-low surface tension, and strong permeability. SC–CO_2_ injection into shale changes the pore characteristics of shale, reduces the specific surface area, increases the porosity and average pore size, and improves the fracturing effect [4]. SC–CO_2_ fracturing technology can solve the shortcomings of water-based fracturing fluid systems such as large waste of water resources, clay swelling, and residual working fluid that cause damage to the reservoir and incomplete reverse drainage to cause groundwater pollution [5,6,7].

According to existed research, the advantages of fluoropolymers [8] in chemistry and low surface tension make them widely used in the area of oil field chemistry. The reported fluoropolymers [9,10,11] generally have an excellent performance in SC–CO_2_ thickening, but fluorine-containing monomers are expensive. Fluorine was widely used in aerospace, petroleum, chemical, and other areas. It flows into the water cycle through wastewater discharge, and it cannot be metabolized and is enriched in biological extraction, causing great harm [12]. The reported siloxane thickeners [13,14,15,16,17] rely on the formation of Lewis acid hydrogen bonds between the cosolvent and CO_2_ and the similar compatibility of the cosolvent with siloxane to enhance solubility. Silicone has the characteristics of the low glass transition temperature, low cohesive energy, and good economy. Modified silicone thickeners have great development potential. The reported hydrocarbon thickener [18,19]: (1) Low-molecular-weight compounds have good solubility in CO_2_ and poor thickening effect, (2) Long-chain polymer hydrocarbons are soluble in CO_2_ under co-solvent conditions, but the damage of co-solvents to the formation cannot be ignored [14]. In this work, we designed and prepared two environmentally friendly supercritical carbon dioxide thickeners, which solved the problem of high cost and high pollution caused by the use of fluoropolymer and cosolvent in the past. Moreover, the thickener has excellent solubility and thickening performance and has certain temperature resistance and pressure resistance.

A self-designed high-precision capillary viscometer was used to measure the apparent viscosity of the CO_2_ fracturing fluid after thickening; the capillary viscometer was used in conjunction with a core displacement device to study core damage and fluid loss. In addition, the contents of different thickeners, the influence of temperature, pressure, and concentration on CO_2_ viscosity, combined with Materials Studio and Abaqus to study the solubility, thickening mechanism, and fracturing simulation respectively.

## 2. Materials and Methods

### 2.1. Materials Studio Simulation

The polymer simulated in this paper was named HBD-1 and HBD-2, and a CO_2_ system with 1000 CO_2_ molecules, a polymer system with 4 polymer chains, and a polymer with 4 polymer chains 1000 CO_2_ molecules were established by using the Material Studio software (Accelrys Ltd., San Diego, CA, USA), the all-atom molecular model of CO_2_ system, as shown in Figure 2.

After the Forcite optimization and annealing calculation of the amorphous cell module, MD simulation of the amorphous cell module was carried out in the NPT system. The temperature was set to 305.15 K, and the temperature was set to 0.01 GPa. The running time for all the systems was 400 ps, and the trajectories were saved at 1 ps intervals [20,21,22].

### 2.2. Synthesis and Characterization of the Hyperbranched D4H

The materials used in the experiment were ethylene glycol dimethacrylate, 2,4,6,8-tetramethylcyclotetrasiloxane (D4H), trimethylolpropane trimethacrylate and H_2_PtCl_6_·6H_2_O, which were purchased from Macleans. HBD-1 and HBD-2 were synthesized by hydrosilylation [23] according to the procedure shown in Scheme 1.

Ethylene glycol dimethacrylate was washed with 1–5% NaOH alkali solution for three times to remove MEHQ, and then washed for three times to remove NaOH. Anhydrous magnesium sulfate was dried, filtered to remove magnesium sulfate, distilled to remove the remaining alkali, and then refrigerated at low temperatures. First, 16.4833 g of ethylene glycol dimethacrylate was added to a 250 mL three-necked flask under magnetic stirring. When the solution temperature rises to 70 °C, 40 ppm catalyst was added to activate for 2 h. Then 10 g of D4H (2,4,6,8-tetramethylcyclotetrasiloxane) was slowly dropped into a three-mouth bottle (about 30 min~35 min), and HBD-1 initial product was obtained after 4 h reaction. To remove chloroplatinic acid, 1 g activated carbon was added into the product, and the mixture was repeatedly washed with distilled water after hydrosilylation was stopped. Under the vacuum conditions of 370 K and 0.06 MPa, the small molecule compounds and water were removed by the rotating evaporator. At 90 °C, the ethylene glycol dimethacrylate was replaced by 16.8841 g trimethylolpropane trimethacrylate to produce HBD-2.

#### 2.2.1. H NMR Measurements

Bruker-400 MHz NMR (Bruker Ltd., Switzerland) was used to characterize the ^1^H NMR spectrum of the polymer with deuterated chloroform as solvent [20,24,25].

#### 2.2.2. FT-IR Measurements

The infrared spectrum measurement of the polymer was completed by Nicolet iS50 Fourier transforms infrared spectrometer (Thermo Fisher Scientific Ltd., Waltham, MA, USA). After washing the ATR crystal with ethanol, spread 0.10 mL of the sample evenly on the surface of the ATR. The scan number of the spectrum is 128 and the resolution is 8 cm^−1^. The wave number ranges from 4000 to 400 cm^−1^ [26].

#### 2.2.3. GPC Measurements

The measurement of polymer molecular weight is done by WATERS 2414 refractive index detector (Waters Ltd., Shanghai, China). Prepare 5 mg/mL polymer solution with chromatographic grade tetrahydrofuran and pass the solution through the gel chromatography column at a flow rate of 1 mL/min at 35 °C [27].

#### 2.2.4. Differential Scanning Calorimetry

The glass transition temperature of the polymer was measured using a Mettler-Toledo differential scanning calorimeter in nitrogen atmosphere (Mettler-Toledo Co., Shanghai, China). A 10 mg polymer sample was placed on the bottom of an aluminum crucible and sealed with a porous lid. The heating rate and cooling rate are 5 °C/min. Take the average of three measurements as the result [28,29,30].

#### 2.2.5. Viscosity Measurement

The capillary viscometer (Figure 3) was composed of a plunger pump, a visualization chamber, a viscosity measurement part, and a data collection terminal to study the viscosity of SC–CO_2_ fracturing fluid after thickening. The inner diameter of the capillary is D = 0.8 mm First, carbon dioxide and thickener are pumped into intermediate container Ⅰ through booster pump I, booster pump II, and the second ISCO pump pumps the mixed thickener and CO_2_ into the visualization container, turn on the heating area to ensure that the conditions in the container reach T = 304.25 K, P = 7.38 MPa [31] (supercritical conditions) or above, and then the clarified mixed liquid was pumped into the intermediate container II and pressed into the capillary at a constant flow rate In the end, the pressure difference recorded by the differential pressure sensor at both ends of the capillary are expressed in Equation (1), which was generally suitable for laminar fluids [32].
(1)$$\eta \;=\;\frac{\tau_{w}}{\gamma_{w}}\;=\;\frac{D∆p/4L}{8v/D}$$
where η was fluid apparent viscosity (Pa s), τ_w_ was wall shear stress (Pa), γ_w_ was the apparent shear rate (s^−1^), D presented the capillary diameter (m), Δp was the pressure difference of capillary (MPa), and L was capillary length (m), v was the flow velocity of thickened liquid CO_2_ (m s^−1^).

#### 2.2.6. Cloud Point Measurement

Figure 4 was used for cloud point measurement. Before the experiment, open switch 5 to drain the air in the visualization container by a vacuum pump (the switches not mentioned in each step were closed), and the left side of the intermediate container II was filled with water. (1) 1 wt.% HBD thickener (305.15 K) was added to the visualization reactor. (2) Turn on the 1,2,6 switches, compress the CO_2_ into liquid in the intermediate container I with ISCO pump, turn on switch 4 to transfer the liquid CO_2_ to the visible container, and operate repeatedly until the container was filled. (3) After the thickener was mixed with CO_2_, turn on switches 3 and 4, and slowly draw out the homogeneous liquid in the visible container with a hand pump to observe the cloud point phenomenon (the pressure when the homogeneous solution was turbid). In the whole experiment process, the pressure rise was controlled by ISCO pump, and the depressurization was controlled by hand pump.

#### 2.2.7. Core Damage Measurements

Before the start of the experiment, pure SC–CO_2_ was injected into the core holder (Figure 5) at an injection rate of 0.181 mL/min, the pressure difference between the two sides of the core holder was recorded, and the permeability of the heterogeneous core was calculated, Equation (2). In this work, the displacement fluid is supercritical carbon dioxide. The results show that SC–CO_2_ has no damage to the core studied. After thickening, the SC–CO_2_ displacement experiment was carried out at 0.18 mL/min. The specific experimental steps are as follows: (1) The initial permeability was measured by vacuum pumping and supercritical carbon dioxide saturation. (2) At a constant flow rate of 0.18 mL/min, the thickened supercritical carbon dioxide was injected to keep the fluid pressure stable in the intermediate vessel. (3) During the whole experiment, the change of injection end pressure was monitored, the backpressure was kept at 8 MPa, and the confining pressure was always 4 MPa higher than the inlet pressure. The pressure difference between the two sides of the core holder was recorded, and the permeability of the heterogeneous core after fracturing was calculated.
(2)$$k=\frac{q\mu L}{A∆p}$$
where k was permeability (μm^2^), *q* was flow rate (cm^3^/s), *μ* was the viscosity (mPa·s), *L* presented the core length (cm), *A* was core cross-sectional area (cm^2^), and Δ*p* was the pressure difference of core holder (mPa).

## 3. Results and Discussion

### 3.1. Materials Studio Computational Simulation

#### 3.1.1. Interaction Energy Calculation

In the polymer-CO_2_ system, the dissolution of the polymer in CO_2_ mainly depends on the polymer-CO_2_ interaction. The interaction energy (E_inter_) can quantitatively characterize the intensity of its action. The greater the absolute value of the interaction energy was, the stronger the polymer-CO_2_ interaction was. First, calculate the total energy E_polymer-CO2_ of the polymer chain-CO_2_ in the above stabilization, then calculate the E_polymer_ and E_CO2_(The calculation results are recorded in Table 1), and finally the interaction energy was calculated by Equation (3) [20].
E_inter_ = E_polymer-CO_2__ − (E_polymer_ + E_CO_2__)(3)

#### 3.1.2. Cohesive Energy Density and Solubility Parameter

The cohesive energy density (CED) and solubility parameters are also generally used to represent the interaction between polymer molecules. The cohesive energy density was the energy required for vaporization of 1 mol condensate per unit volume to overcome the intermolecular force, and mainly reflects the interaction between groups. The square of the solubility parameter was the cohesive energy density. Relevant studies have shown that polymers with lower cohesive energy density have a higher solubility in CO_2_ [33,34,35], and the smaller the difference between the solubility parameter of the polymer and CO_2_ was, the better the solubility of the polymer in CO_2_ was [36]. The CED and solubility parameter (δ) of the polymer and CO_2_ was shown in Table 2. The |∆δ| of the two systems in Table 2 are 1.08 and 0.04 respectively.It can be seen that HBD-1 and HBD-2 have good solubility in SC–CO_2_.

#### 3.1.3. Radial Distribution Function (RDF)

The mechanism of polymer solvation in SC–CO_2_ was studied by comparing the RDF value of carbon atom in polymer-CO_2_ with that of carbon atom in polymer-polymer. If the RDF value of polymer-CO_2_ was larger, it proves that polymer HBD-1 was miscible with SC–CO_2_ [20,21,22,37,38]. The results of the polymer-CO_2_ and polymer-polymer are shown in Figure 6. The results show that the RDF value of polymer-CO_2_ was greater than that of polymer-polymer, and polymer HBD-1and HBD-2 has good miscibility in SC–CO_2_.

### 3.2. Structural Characterization of HBD-1 and HBD-2

The FT-IR and ^1^H NMR spectra of D4H, HBD-1, and HBD-2 were recorded in Figure 7 and Figure 8. In the infrared spectra of D4H, HBD-1, and HBD-2, there were Si—O characteristic peaks at 1050 cm^−1^~1100 cm^−1^ [15], and Si—C characteristic peaks at 850 cm^−1^~890 cm^−1^. It can be seen from Figure 8 that the Si—H peak at 2160 cm^−1^ was significantly weakened, and a C=O peak appeared at 1735 cm^−1^ [39]. In the FT-IR of HBD-2, a peak of 1635 cm^−1^ appeared. The numerical changes of these characteristic peaks indicate the changes in the functional groups corresponding to the hydrosilylation reaction. The double bond peak still exists in the infrared image of HBD-2 because the double bond is excessive in the reaction.

The ^1^H NMR shift data of HBD-1 and HBD-2 were shown in Table 3. It can be seen from Figure 7 that the Si—H peak in D4H appears at 4.5 ppm. In combination with Table 3, Figure 7 and Figure 8, it can be seen that the chemical shift in the ^1^H NMR spectrum indicates that the two polymers undergo addition reactions with two symmetrical H in D4H, so Si—H still exists in HBD-1 and HBD-2. However, there are double-bonded hydrogens at 6.02 ppm~6.4 ppm in the HBD-2 ^1^H NMR spectrum. This is because the monomer trimethylolpropane trimethacrylate that participates in the hydrosilylation reaction contains excessive double bonds, and these double bonds do not participate in the reaction due to steric hindrance. The numerical changes of these characteristic peaks indicate the changes in the functional groups corresponding to the hydrosilylation reaction. The double bond peak still exists in the infrared image of HBD-2 because the double bond is excessive in the reaction. Combined with FT-IR and ^1^H NMR, HBD-1 and HBD-2 were successfully prepared.

### 3.3. Glass Transition Temperature and Molecular Weight Analysis

Figure 9a,b are the DSC and GPC spectra of HBD, respectively. It can be seen from (a) that glass transition temperature (T_g_) of HBD-1 and HBD-2 is −15 °C and −44.5 °C, respectively. The T_g_ was generally used to characterize the relative flexibility of polymer chains, the lower the glass transition temperature was, the better the flexibility of the molecular chain was [40,41]. High polymer chain flexibility helps to dissolve HBD in SC–CO_2_ [35]. The polymer parameters were shown in Table 4.

### 3.4. Cloud Point and Viscosity of HBD-1 and HBD-2 in SC–CO_2_

#### 3.4.1. Cloud Point and Phase Behavior of HBD-1 and HBD-2 in SC–CO_2_

The phase behavior system consists of thickener HBD and SC–CO_2_. It can be seen that pure CO_2_ was a transparent liquid (298.15 K, 7.48 MPa). In Figure 10b, HBD-1 was milky white and HBD-2 was translucent, which indicates that HBD-2 (298.15 K, 7.48 MPa) was more soluble in SC–CO_2_. The experimental results of Figure 10c show that HBD-1 and HBD-2 thickeners have excellent solubility in SC–CO_2_ after being kept at a constant temperature of 305.15 K for 12 h. This was in good agreement with the results of E_inter_, CED, and RDF. As well, the cloud point pressure of the thickener SC–CO_2_ system in this study was lower than 7.48 MPa.

The mechanism was to introduce multiple aliphatic groups and an octane cyclosiloxane into the molecular design of thickeners. Carbon dioxide and lipid groups form hydrogen bonds through Lewis acid-base pairs [42,43,44]. The good chain flexibility of siloxane also plays a role in improving the solubility of the polymer in SC–CO_2_ [35,45].

#### 3.4.2. The Influence of Temperature on the Apparent Viscosity of HBD-1 and HBD-2 in SC–CO_2_

The effect of temperature on the apparent viscosity of thickened SC–CO_2_ was recorded in Figure 11. The results show that the apparent viscosity of the solution decreases with the increase in temperature. The results are the same as the previous studies on the effect of temperature on the apparent viscosity, because the temperature has a certain influence on the network structure of polymer chains [46,47,48]. The results also show that HBD-2 thickener has better temperature resistance when the temperature rises from 305.15 K to 315.15 K. This was because there are branched chains in HBD-2 polymer chains, and the polymer network structure can still maintain a certain tight network structure of the temperature action.

The mechanism was that when HBD polymer molecules enter SC–CO_2_, the molecular chains entangle with each other in the form of curl, forming a complex three-dimensional network structure. The molecular structure contains an eight-membered ring, and there are hydrogen bonds (carbonyl and CO_2_) in the system [42,43,44], which hinder the flow of CO_2_ molecules. When the initial temperature was low, the apparent viscosity decreased slowly. With the increase in temperature, the network structure was destroyed more and more thoroughly, and the apparent viscosity decreased sharply.

#### 3.4.3. The Influence of Pressure on the Apparent Viscosity of HBD-1 and HBD-2 in SC–CO_2_

Figure 12 records the response of pressure to the apparent viscosity of thickened SC–CO_2_. The results show that the apparent viscosity of SC–CO_2_ increases slowly from the increase in system pressure. According to the thickening mechanism, the intermolecular distance decreases to the increase in pressure, which makes it easier to form a more compact three-dimensional network structure [47]. In the process of pressurization, the self-winding of the polymer chain becomes closer, and the hydrogen bonding between the electron-donating group and CO_2_ (Lewis acid-base formation) gradually increased [49], which was conducive to the increase in apparent viscosity. It can be seen from the study in Figure 12 that the viscosity increasing effect of HBD-2 was better than that of HBD-1 in the range of 7.48 MPa~14 MPa. It can be seen from Figure 12 that HBD-2 is more sensitive to pressure in the same pressure range. This is due to the existence of branched chains in the polymerization unit of HBD-2, and the network structure formed between the polymer molecules is more compact than that of HBD-1 after increasing the pressure. The denser network structure restricted the free flow of CO_2_, and the viscosity of SC–CO_2_ thickened by HBD-2 increased more obviously under the same pressure change. It was pointed out that the molecular design of thickeners can enrich branch chains in a certain range.

#### 3.4.4. The Influence of Shear Rate on the Apparent Viscosity of HBD-1 and HBD-2 in SC–CO_2_

Figure 13 shows the change in the apparent viscosity of SC–CO_2_ after thickening in the shear rate range of 60 s^−1^ to 120 s^−1^. Firstly, it can be seen from the figure that the change of shear rate does not affect the viscosity of pure CO_2_, because pure CO_2_ is a Newtonian fluid. HBD-1 and HBD-2 exhibit a negative correlation with shear rate, which proves that SC–CO_2_ fluid after thickening is a power-law fluid with shear-thinning [43]. After thickening, the apparent viscosity of SC–CO_2_ fluid decreases with the increase of shear rate, mainly because the increase of shear rate destroys the spatial network structure and restores the fluidity of CO_2_ molecules that were originally restricted to flow.

#### 3.4.5. The Influence of Thickener Content on the Apparent Viscosity of HBD-1 and HBD-2 in SC–CO_2_

Figure 14 shows the effect of thickener content on the apparent viscosity of thickened SC–CO_2_. The results show that the apparent viscosity increases in the range of 1 wt.%~5 wt.%. The mechanism of action was the number of polymer molecular chains that can form a tight network structure and hydrogen bonds in the system increased exponentially. Destroying these structures requires a lot of energy [50], and there is no external energy under experimental conditions. Therefore, the free-flowing SC–CO_2_ molecules in the system were captured by the expanding polymer network, which shows the increase in apparent viscosity.

### 3.5. Core Damage

In the development of oil and gas resources, there must be physical, biological, and thermal interactions between formation and fluid [51]. Air and water pollution caused by traditional hydraulic fracturing were expected to be solved by SC–CO_2_ water-free fracturing technology. In this study, a blank test was conducted first. The initial permeability measurement results are shown in Table 5.

The results show that the permeability of the core saturated with SC–CO_2_ was the same as that of the core without SC–CO_2_ immersion. Formation damage rates Equation (4) is shown as following:(4)$$\varphi \;=\;\frac{K_{1}{\;-\;K}_{2}}{K_{1}}\;\times \;100\%$$
where φ is formation damage rate, K_1_ is matrix permeability before fracturing fluid injection (×10^−3^ μm^2^), K_2_ is matrix permeability after fracturing fluid injection (×10^−3^ μm^2^).

Figure 15 shows the permeability loss rates of three cores with different permeability after fracturing with 5 wt.% HBD-1 and HBD-2 thickened SC–CO_2_. The results show that each core has a certain degree of permeability loss, which shows that the core with higher permeability has less permeability loss and vice versa. This may be due to the fact that some thickeners adsorbed on the rock surface change the wettability, which leads to a decrease of permeability [52].

## 4. Conclusions

The SC–CO_2_ thickener molecules HBD-1 and HBD-2 were designed by molecular simulation method, and the interaction energy, CED, and RDF of the thickener-CO_2_ system were calculated. Using a self-designed and assembled capillary viscometer, the effects of content, temperature, shear rate, and pressure on the apparent viscosity of SC–CO_2_ were studied. The excellent solubility of HBD-1 and HBD-2 thickeners was studied through simulation experiments and visualization experiments. The capillary viscometer was combined with a core displacement device to study the damage of thickened SC–CO_2_ to the core. The results show that HBD-1 and HBD-2 (305.15 K, 7.48 MPa) have good solubility in SC–CO_2_; the apparent viscosity of supercritical CO_2_ fluid after thickening is positively related to pressure and dosage and is related to temperature and deceleration rate Negative correlation. HBD-2 has a better thickening effect at 5 wt.%, and the apparent viscosity can reach 4.48 mPa·s (305.15 K, 10 MPa).

There are three main mechanisms for polymer thickening and dissolution in SC–CO_2_: polymer-polymer interaction, polymer-carbon dioxide interaction, and polymer chain flexibility. To avoid the use of fluoropolymers and co-solvents, following three mechanisms, the author introduced some CO_2_-philic groups in the polymerization unit to increase the solubility of polymer molecules and introduced cyclic siloxanes to improve the viscosity-increasing effect. A thickener with good solubility in supercritical CO_2_ was designed. This study verified the feasibility of this idea and found that the existence of multiple aliphatic groups in the branched-chain also had a certain impact on the viscosity, which opened up a new way for the development of environmentally friendly SC–CO_2_ thickener.

## Data Availability

The data presented in this study are available in the insert article.

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
