# Peer review of "Preparation and Performance of Supercritical Carbon Dioxide Thickener"

_polymers, 2020, doi:10.3390/polym13010078_

Round 1

Reviewer 1 Report

In this work, two synthetized structures were characterized by H NMR, FT-IR, GPC and DSC. Viscosity measurements were performed with a self-designed and assembled device, and the damage of SC-CO2 fracturing fluid to formation was studied. The work is interesting and can be considered as a potential application in the industry. However, before publication, the following comments must be heeded.   1. The manuscript reports data on different experimental measurements, however, the uncertainty analysis is not reported. For each measurement carried out, the analysis must be performed, in order to know the reliability of the experimental data.   2. The equation presented for the determination of viscosities does not consider effects such as kinetic energy or end effects. What was the approach for not considering these effects? What are the dimensions of the capillary?   3. Figure 12 shows the effect of pressure on viscosity. Much greater effect is seen in HBD-2 but the reason is not explained. Authors should explain the most noticeable effects in all Figures presented.   4. In Figure 14, the HBD-2 structure shows an increase in the rate of loss of permeability but in the end, the rate of loss decreases (13-15-14), why this effect?   5. The authors say "The results show that HBD-1 and HBD-2 have excellent solubility in SC-CO2". However, there is no appreciable support throughout the manuscript. Section 3.1.2 (Cohesive energy density and solubility parameter), should improve considerably, in order to appreciate and verify the excellent solubility that the authors mention.   As mentioned above, I recommend that the manuscript be published after major corrections noted.

Reviewer 2 Report

In this paper authors present a theoretical and experimental study of SC-CO2 thickener.  The results presented in the manuscript seems to be novel. The experimental approach is adequately presented in the text of the manuscript. However, the quality of presentation of numerical modeling should be improved before publication. The manuscript is well organized and seems interesting to a wide range of auditory. It could be accepted after some changes.

  1. The procedure of simulation description of numerical simulation should be expanded.
    1. What interatomic potential was used in the simulation?
    2. Can the potentials be applied for the system?
    3. Does numeric simulation adequately reproduce the macroscopic properties of the system (heat capacity, etc.)?
    4. How was the running time chosen? Is it enough for equilibration of the system?
    5. How was exactly the CED calculated?
    6. In the Fig.6 the author shows the behavior of the RDF of C-C pairs for polymer-CO2 system. However, it is difficult to estimate the adequacy of this data. Can authors present the C-C RDF for pure CO2?
  2. The quality of the Fig. 10 does not make it possible to compare the phase behavior of the medium with the description in the text of the manuscript.
  3. The authors work slightly above the critical point. However, the critical point of the mixture differs from the critical point of pure CO2. Thereby, is it correctly to claim that the medium (CO2+HBD) is supercritical?

In addition, there several minor issues that should be corrected.

  1. Pressure is not indicated in Fig.11 and temperature is not indicated in the Fig.12.
  2. Do the authors estimate the viscosity of the medium in the sub-critical region?

Reviewer 3 Report

Manuscript ID; polymers-1038498   Title; Preparation and Performance of Supercritical Carbon Dioxide Thickener Regarding the submitted paper to Polymers Journal, the proposed topic is interesting and would be applicable for industrial purposes, however, it needs some modifications prior to publication. My major concerns are as follows; 1- In the Introduction section, it is necessary to discuss more what are the differences between your paper with previous papers. For example, you should say, .....et al() did this and we do this and tell why your work is important. Moreover, supercritical properties of CO2 should be discussed and focused. What are the limitations????????????? 2- In the introduction section, the following references about shale formations should be cited and discussed. - Thermodynamic effects of cycling carbon dioxide injectivity in shale reservoirs - Using Photo-Fenton and Floatation Techniques for the Sustainable Management of Flow-Back Produced Water Reuse in Shale Reservoirs Exploration - Hybrid Thermal-Chemical Enhanced Oil Recovery Methods; An Experimental Study for Tight Reservoirs - Experimental investigation and mathematical modeling of gas diffusivity by carbon dioxide and methane kinetic adsorption - The feasible visual laboratory investigation of formate fluids on the rheological properties of a shale formation -A feasible visual investigation for associative foam\polymer injectivity performances in the oil recovery enhancement   3- Change section 2 to "Materials and Methods" and reorganize this section again. Moreover, you should explain the Synthesis method in more detail and how do you do this in the lab?   4- Page 5 line 125, The solubility of the thickener in the SC-CO2 system increases with pressure. Please elaborate more????   5- In figure 5, what is the value of confining pressure and back pressure??????   6- Section 2.2.7, how do you measure core damage, and can you explain more about the reliability of your experiments???????   7- Figure 7 and 8 should be explained more   8- Can you discuss what was happened in high-pressure high-temperature wells?????? Does your new material stabilize in HP-HT condition???   9-Could you please discuss polymer degradation, shear thickening rate, and other properties of the polymer in more detail?
  9- Abstract and conclusion should be improved and write more scientifically   11- A nomenclature section should be added to define each parameter and unit.
